# Highly atroposelective synthesis of nonbiaryl naphthalene-1,2-diamine N-C atropisomers through direct enantioselective C-H amination

He-Yuan Bai [1], Fu-Xin Tan[1], Tuan-Qing Liu[1], Guo-Dong Zhu[1], Jin-Miao Tian[1], Tong-Mei Ding[1], Zhi-Min Chen[1] & Shu-Yu Zhang [1]

Nonbiaryl N-C atropisomer is an important structural scaffold, which is present in natural products, medicines and chiral ligands. However the direct enantioselective C-H amination to access optically pure N-C atropisomer is still difficult and rare. Here we report a π-π interaction and dual H-bond concerted control strategy to develop the chiral phosphoric acids (CPAs) catalyzed direct intermolecular enantioselective C-H amination of N-aryl-2-naphthylamines with azodicarboxylates as amino sources for the construction of atroposelective naphthalene-1,2-diamines. This type of N-C atropisomers is stabilized by intramolecular hydrogen bond and the method features a broad range of substrates, high yields and ee values, providing a strategy to chirality transfer via the modification of N-C atropisomers.

---

[1] Shanghai Key Laboratory for Molecular Engineer of Chiral Drugs & School of Chemistry and Chemical Engineering, Shanghai Jiao Tong University, Shanghai 200240, China. Correspondence and requests for materials should be addressed to S.-Y.Z. (email: zhangsy16@sjtu.edu.cn)

Atropisomeric compounds, in which the chirality generates from a chiral axis with highly sterically hindered rotation, are among the most useful catalysts or ligands in enantioselective catalysis and have been attracting considerable attention of chemists[1–6]. To date, the biaryl atropisomers, such as BINAP[7,8] and BINOL[9,10], have been well developed and widely used. In sharp contrast, the catalytic atroposelective construction of nonbiaryl atropisomers, which is of critical importance as medicine or chiral ligands in our lives (Fig. 1a)[11–14], due to the rotational restriction around an N-C single bond, remains largely unexplored. Until recently, the preparation of these optically active N-C nonbiaryl atropisomers still depends largely on the chiral resolution and diastereoselective synthesis using chiral pool precursors[15–17]. Only few strategies for catalytic atroposelective construction of N-C nonbiaryl atropisomers exist and mainly consist of enantioselective cyclization[18–20], N-functionalization[21–27], and desymmetrization[28–30] of the existing achiral N-C bond. The direct catalytic enantioselective method to build a new chiral N-C bond to form nonbiaryl N-C atropisomers still has significant synthetic challenges and has attracted great interest.

Over the past three years, our laboratory has been engaged in developing efficient C-H amination methods[31–34]. Among them, azodicarboxylates, as a special amino source, have attracted our great interest. In the course of our investigation, we found that 2-naphthylamine derivatives could succeed in constructing C-N bonds at C1 position with azodicarboxylates catalyzed by a transition metal or Brønsted acid. Inspired by these results on the C1 position of C(sp²)-H amination, we proceeded to investigate whether the challenging nonbiaryl axially chiral N-C bonds could be formed by this direct C-H enantioselective amination, which would be an arguably ideal and attractive approach with great atom and step economy.

The previous landmark study by Jørgensen's group[35] introduced the enantioselective nonbiaryl axially chiral amination of 8-amino-2-naphthol derivatives with azodicarboxylates catalyzed by cinchona alkaloids (Fig. 1b). This study indicated that the amino group of C-8 position played an indispensable role in the stability of axially chirality and therefore the amination was limited by substrate scope. In 2014, Gong's group[36] reported the chiral Au-catalyzed cycloisomerization amination cascade reaction with azodicarboxylates to build heteroaryl atropisomers. Much disclosed in this context that the nonbiaryl atropisomers of diazenes were easily racemized and the single crystal of optically pure heteroaryl atropisomers was unable to be obtained. Despite these limits and challenges, it provided the foundation for our process.

Herein, we design a π–π interaction and dual H-bond concerted control strategy to develop the chiral phosphoric acids (CPAs) catalyzed direct intermolecular enantioselective C-H amination of N-aryl-2-naphthylamines with azodicarboxylates for the construction of N-C atroposelective nonbiaryl naphthalene-1,2-diamines (Fig. 1c).

## Results

**Reaction strategy.** Our initial strategy was inspired by the pioneering studies of Akiyama[37] and Terada[38] on the use of CPAs as organocatalysts, which were widely used in organic synthesis. In line with our recent interest in the C-H amination reaction and the use of azodicarboxylates as an amino source, we believe that 2-naphthylamine as a nucleophiles derivative has a weakly acidic hydrogen atom, which could be activated and form a dual hydrogen-bonding intermediate **A** with azodicarboxylates under CPA as a catalyst to simultaneously activate two reaction partners and thus construct a chiral N-C axis (Fig. 2a). It should be mentioned, while chiral 2-naphthylamine derivatives such as BINAP and NOBIN[39–41] are widely used as enantioselective catalysts in various reactions, the enantioselective construction of 2-naphthylamine derivatives is still rare[42–47]. Recently, Tan[48] reported the organocatalytic arylations of 2-naphthylamine to successfully form biaryl atropisomers. However, the more

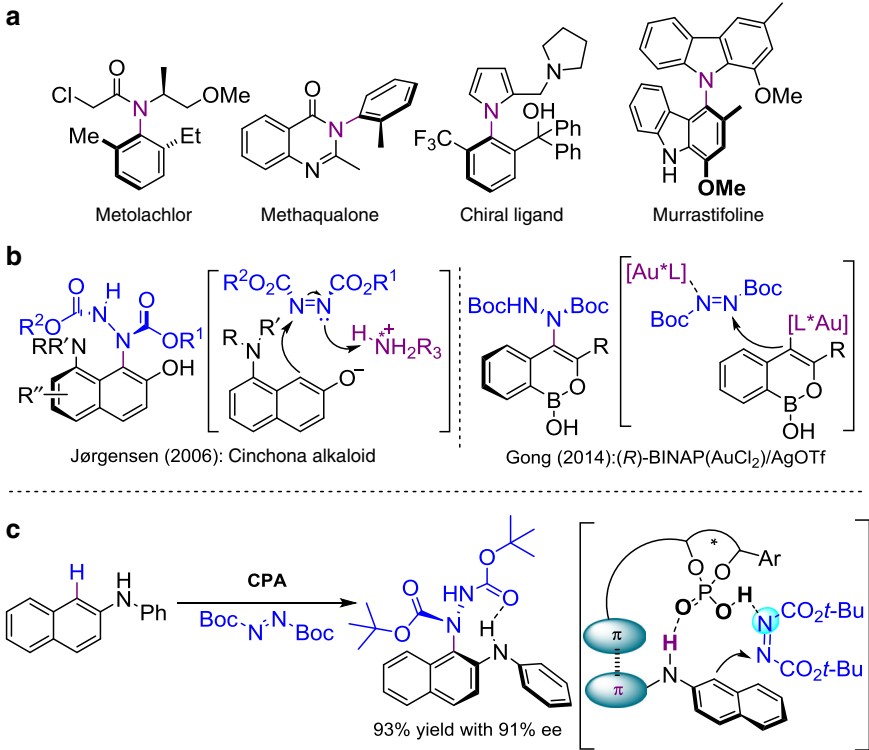

**Fig. 1** N-C nonbiaryl axially chiral structure. **a** N-C axially chiral compounds. **b** Previous N-C atropisomers with azodicarboxylates. **c** This work: CPA-catalyzed direct atroposelective C-H aminations via a concerted control of π-π interaction and dual H-bond strategy

challenging chiral nonbiaryl N-C atropisomers of 2-naphthylamine derivatives have not been discovered.

Based on this initial strategy mentioned above, we started to evaluate our hypothesis by using N-substituted 2-naphthylamine **1** and di-*tert*-butyl azodicarboxylate (DBAD) **2a** as model substrates. As shown in Table 1, when the substrate **1a** (with hydrogen atom substituted by methyl and phenyl) was used, the reaction failed to generate any desired product using **CPA1-3** as catalysts (Table 1, entries 1–3). Interestingly, when **1b** and **1c** (the substrate with an N-H bond) were used in the amination (Table 1, entries 4–7), the amination underwent smoothly and gave the

amination products in 45–74% yields. These results showed that the N-H bond was indispensable and played a critical role in this process, which was in agreement with our initial design. However, the enantioselectivity of the desired product was very low (4–8% ee) even when the more sterically hindered Ph- or Ph₃Si-substituted CPAs **CPA2** and **CPA3** were used. As shown in Fig. 2a, we believe the main reason is that the rotation is unhindered using only the dual H-bond to control the transition state of intermediate **B**. While the general approach to install some large groups at 2-naphylamine substrates (C8 position) might limit the rotation of intermediate **B** and partially probably improve the stereoselective control, however, it will severely limit the versatility and scope of substrates and applications of this chiral amination reaction.

To overcome these drawbacks and limitations, we designed a weak intermolecular interaction to help limit the contortion of intermediate **B** and enhance the stereoselectivity control of our C-H amination reaction. The π-π interaction[49,50] as an arguably ideal modified strategy was introduced in our enantioselective axially chiral amination (Fig. 2b, c). To evaluate the efficiency of our modified design, we used N-phenyl substituted 2-naphthylamine **1d** to react with DBAD in the presence of phenyl-substituted **CPA3** as a catalyst (Table 1, entry 8). Remarkably, the enantioselectivity of this N-C axially chiral amination reaction was significantly promoted and gave the 47% ee with an isolated yield of 45%. We also investigated some more protecting groups of the nitrogen with the more steric hindrance, such as benzyl (**1e**), *tert*-butyl (**1g**), and they all gave low enantioselectivity (Table 1, entries 9–11). These results strongly support our modified design in Fig. 2b, and clearly indicate that the concerted control strategy of π-π interaction and dual H-bond

**Fig. 2** Reaction strategy. **a** Initial dual hydrogen-bonding strategy. **b** Modified strategy: concerted control of π-π interaction and dual H-bond strategy

**Table 1 Screening results of reaction strategy**

**1a**; R¹= Me, R²=Ph
**1b**: R¹= H, R²= H
**1c**: R¹= H, R²= Me
**1d**: R¹= H, R²= Ph
**1e**: R¹= H, R²= Bn
**1f**: R¹= H, R²= *t*-Bu
**1g**: R¹= H, R²= Allyl

**CPA1**, R = H
**CPA2**, R = SiPh₃
**CPA3**, R = Ph

| Entry | 1 | Catalyst | Dual H-Bond | π-π Interactions | 3 Yield[a] ee[b] |
|---|---|---|---|---|---|
| 1 | **1a** | **CPA1** | No | No | No reaction |
| 2 | **1a** | **CPA2** | No | No | No reaction |
| 3 | **1a** | **CPA3** | No | Yes | No reaction |
| 4 | **1b** | **CPA3** | Yes | No | 63%, 4% ee |
| 5 | **1c** | **CPA1** | Yes | No | 74%, 7% ee |
| 6 | **1c** | **CPA2** | Yes | No | 71%, 4% ee |
| 7 | **1c** | **CPA3** | Yes | No | 45%, 8% ee |
| 8 | **1d** | **CPA3** | Yes | Yes | 45%, 47% ee |
| 9 | **1e** | **CPA3** | Yes | No | 19%, 9% ee |
| 10 | **1f** | **CPA3** | Yes | No | 23%, 0% ee |
| 11 | **1g** | **CPA3** | Yes | No | 37%, 4% ee |

All screening reactions were carried out in a 10 mL glass vial with a PTFE-lined cap on a 0.1 mmol scale. 2.0 equiv of **2a**, 10 mol% catalyst, 1 mL DCM, 25 °C for 3 h
[a]Yield represents isolated yield
[b]Determined by HPLC analysis

**Table 2 Optimization of reaction conditions**

| Entry | R | Cat | Solvent | Temp/°C | Time/h | Yield %[a] | ee %[b] |
|---|---|---|---|---|---|---|---|
| 1 | Ph | CPA3 | DCM(D) | −60 | 12 | 63 | 73 |
| 2 | Ph | CPA4 | DCM | −60 | 12 | 24 | 62 |
| 3 | Ph | CPA5 | DCM | −60 | 12 | 75 | 74 |
| 4 | Ph | CPA6 | DCM | −60 | 12 | 94 | 74 |
| 5 | Ph | CPA7 | DCM | −60 | 12 | 58 | 73 |
| 6 | Ph | CPA8 | DCM | −60 | 12 | 11 | 12 |
| 7 | Ph | CPA6 | Toluene | −60 | 12 | 24 | 47 |
| 8 | Ph | CPA6 | Et₂O(E) | −60 | 12 | 11 | 41 |
| 9 | Ph | CPA6 | THF | −60 | 12 | <2 | – |
| 10 | Ph | CPA6 | D:E = 1:1 | −60 | 12 | 42 | 88 |
| 11 | Ph | CPA6 | D:E = 7:3 | −60 | 12 | 67 | 90 |
| 12 | Ph | CPA6 | D:E = 7:3 | −70 | 48 | 93 | 91 |
| 13 | Ph | CPA6 | D:E = 7:3 | −78 | 48 | 65 | 92 |
| 14[c] | Ph | CPA6 | D:E = 7:3 | −70 | 48 | 94 | 91 |

All screening reactions were carried out in a 10 mL glass vial with a PTFE-lined cap on a 0.1 mmol scale. 2.0 equiv of **2a**, 10% mol catalyst, 1 mL solvent
[a]Yield represents isolated yield
[b]Determined by HPLC analysis
[c]20 mol% catalyst

between the N-phenyl of 2-naphthylamine and the aryl of CPAs are extremely important and efficient for the control of stereoselectivity in this nonbiaryl axially chiral amination reaction.

**Optimization of reaction conditions**. Based on our final design (Fig. 2b) and considerations mentioned above, we further attempted our chiral amination by using N-phenyl-2-naphthylamine **1d** and DBAD **2a** as model substrates. After an initial screening of the aryl-substituted CPA catalysts (Table 2, entries 1–6), we found that all the aryl-substituted CPAs **CPA3-CPA7** showed good enantioselectivity and the catalyst **CPA6** gave the desired product **3a** with the best result of 94% isolated yield and 74% ee in DCM at −60 °C (Table 2, entry 4). Encouraged by this result, various solvents were screened (Table 2, entries 7–9). A remarkable solvent effect was observed. When the mixed solvent of DCM and Et₂O (1:1) was used as the reaction medium, the product **3a** was notably promoted to 88% ee (Table 2, entry 10). After adjusting the ratio of DCM/Et₂O and changing the temperature, we found that the optimal catalytic system (standard condition) consisted of 2.0 equiv of azodicarboxylate **2a**, 10 mol% of **CPA6** at −70 °C in a mixed solvent (0.1 M, DCM: Et₂O = 7:3) for 48 h, which resulted in 93% isolated yield and 91% ee (Table 2, entry 12).

In addition, lowering the temperature would decrease the chemical yield (Table 2, entry 13) and increasing the amount of catalyst resulted in a little change (Table 2, entry 14).

**Substrate scope**. With the optimized conditions at hand, we first probed the scope of different azodicarboxylate sources

(Table 3b–d). Some commercially available azodicarboxylates, such as DEAD (diethyl azodicarboxylate), DIAD (diisopropyl azodicarboxylate), and dibenzyl azodicarboxylate, could react with **1d** and provide amination products **3b–3d** with high yields. However, it is clear that the enantioselectivity of this axially chiral amination was sensitive to the steric hindrance of azodicarboxylate, and DBAD was still the best azodicarboxylate amino source and gave the desired N-C axially chiral product **3a** in 91% ee. (Table 2, entry 12). Subsequently, the effect of the substituents (R³) on the benzene ring was examined. The electron-donating groups including methyl and methoxy were fully compatible with the reaction condition and gave the axially chiral products **4a** and **4b** in high yields (95% and 92%) with good to excellent enantioselectivities (86% ee and 93% ee). In addition, when the substituent was on meta-position or ortho-position, the reactivity was notably decreased (Table 3, **4d**, **4e**).

Encouraged by these results, we further expanded the scope of functionalized naphthalene rings in order to examine the generality of this axially chiral amination. The N-phenyl-2-naphthylamines substituted with a variety of functional groups, such as alkyl, aryl, halogens, vinyl, and alkynyl, were well tolerated in our reaction system and formed N-C axially chiral products **5a–5p** in good to excellent yields and high enantioselectivities. Using methyl as a substituent at C3 position, the reactivity was notably deceased but the stereoselectivity (**5d**) was slightly affected. The 5, 6, or 7-methyl-substituted substrates, which were away from the reactive site, gave excellent isolated yields of 90–92% and 86–93% ee (Table 3, **5a–5c**). Furthermore, we focused our attention on the amino group at the adjacent C8-position, which was of great importance in Jørgensen's report. When 8-amino substituted N-phenyl-2-naphthylamine was used

**Table 3 Substrate scope**

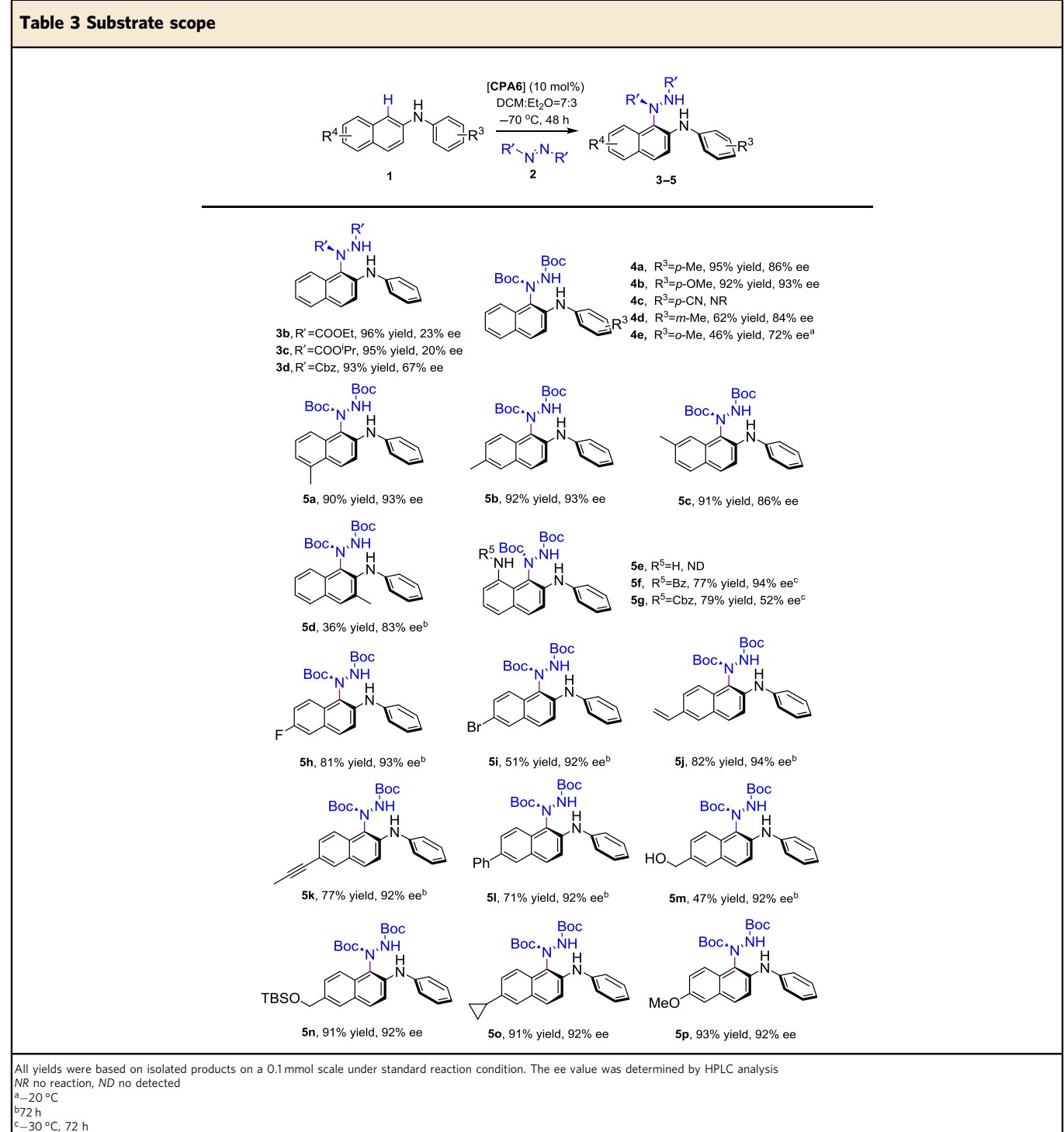

All yields were based on isolated products on a 0.1 mmol scale under standard reaction condition. The ee value was determined by HPLC analysis
*NR* no reaction, *ND* no detected
[a]−20 °C
[b]72 h
[c]−30 °C, 72 h

as a substrate in our reaction conditions, the reaction was very complex and the desired C-1 amination product **5e** was not obtained. To our delight, when Bz was used as an N-protecting group, C-1 regioselective axially chiral amination product **5f** was obtained with 77% yield and 94% ee. The absolute regioselectivity and configuration of **5g** was determined by X-ray crystallographic analysis (see Supplementary Fig. 129 for details).

In addition, various substituent groups were investigated (**5h**–**5p**). The electronic properties of the substituents did not affect the stereoselectivity of the reaction and gave 92–94% ee. The substrates with electron-withdrawing groups would reduce the electron cloud density and limit the reactivity of amination

reaction. For example, the 6-Br substituted N-phenyl-2-naphthylamine gave the desired product **5i** in 51% isolated yield, but with excellent 92% ee.

**Mechanistic study**. Based on our design, experimental results, and literature precedent[51–53], a possible reaction pathway was proposed (Fig. 3). First, the CPA simultaneously activated N-phenyl-2-naphthylamine and azodicarboxylate through a dual hydrogen-bonding activation mode to form intermediate **I**. Then, the π-π interaction assisted intermediate **I** with the concerted control of the enantioselectivity, and the nucleophilic addition of

**Fig. 3** Proposed reaction mechanism. The mechanism was described citing **1d** as an example

N-phenyl-2-naphthylamine to the azodicarboxylate proceeded and formed intermediate **II**. In the following step, rearomatization became favorable for efficient central-to-axial chirality conversion[54] and delivered the desired amination product **3a**. Notably, an intramolecular hydrogen bond (2.233 Å) between the N–H and the oxygen of the carbonyl group was observed and the [1]H NMR chemical shift evidently changed with temperature (see Supplementary Figs. 127–129).

**Stereo-stability study.** To further verify the stereo-stability of our N-C axially chiral products, a series of the half-lives of racemization (in *n*-hexane at 25 °C) of the substituted products were measured. For the substituent group on the benzene ring, the **4b** gave 67.4 h. The substituent groups on the naphthalene ring gave **5a** (103.3 h), **5b** (32.1 h), **5c** (63.3 h), **5 h** (24.8 h), **5i** (46.6 h), **5 l** (52.7 h), **5 m** (33.3 h), and **5p** (20.6 h). The compound **5d** with substituent at adjacent C3 position of naphthalene ring gave the half-life of racemization of 11.4 h. Interestingly, substituent group on C8 position of the naphthalene **5 g** was very stable and gave the half-life of racemization of 5325.2 h. At solid state, we used **3a** (90.10% ee) as an example, and it was 90.04% ee after 90 h at −18 °C. These results indicated that our N-C axially chiral products have suitable stability.

To get a better understanding of the stereo-stability of our N-C nonbiaryl atropisomers, a series of experiments were then carried out to examine the steric hindrance of azodicarboxylates (Fig. 4, Part I), the effect of intramolecular hydrogen bond (Fig. 4, part II), and the substituted groups of the naphthylamine (Fig. 4, part III). Based our above experiment results (**3a–3d**), we knew that the more steric hindrance of azodicarboxylates gave the better enantioselectivity. Furthermore, in order to confirm that the low enantioselectivity was originated from the racemization during the reaction process or the hard stereo-control, we did three parallel experiments to obtain the **3d** at 12 h (64% ee), 24 h (66% ee) and 48 h (67% ee) under standard conditions, showing that the N-C nonbiaryl atropisomers were stable at reaction conditions without racemization. Compared **3a** (92% ee, 70.9 h) and **3d** (67% ee, 25.0 h), this

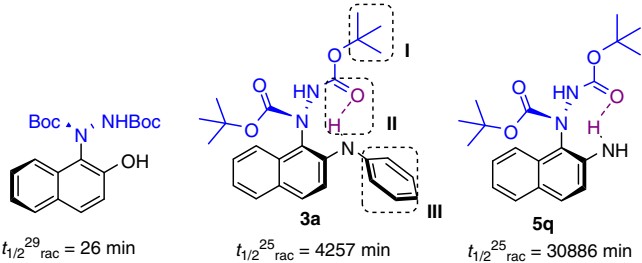

*t*$_{1/2}$$^{29}_{rac}$ = 26 min      *t*$_{1/2}$$^{25}_{rac}$ = 4257 min      *t*$_{1/2}$$^{25}_{rac}$ = 30886 min

**Fig. 4** The Stereo-stability study. The half-lives of racemization of **3a** and **5q** were examined in *n*-hexane at 25 °C

evidence suggested that the large steric hindrance of azodicarboxylates (Fig. 4, part I) not only improved the enantioselectivity of reaction, but also stabilized the N-C atropisomers.

The N-C nonbiaryl atropisomers of our 2-naphthylamine substrates are much more stable than 2-naphthol[35] (Fig. 4), and the intramolecular hydrogen bond[55,56] (conformed by X-ray) would be the important reason. Despite the N-phenyl substituted group was necessary because of π–π interaction in stereo-control in Table 1, the compound **5q** (obtained by chiral resolution), which was much easier to form intramolecular hydrogen bond, was used to compare the half-lives of racemization to examine the intramolecular hydrogen bond effect on stereo-stability and gave 514.8 h. These results indicated that the intramolecular hydrogen bond (N-H⋯O) notably improved the stereo-stability of these N-C axially chiral products.

**Synthetic applicability.** To enhance the practicality and effectiveness of our enantioselective catalytic protocol, a gram-scale reaction was carried out, as shown in Fig. 5a. The desired N-C axially chiral amination product **3a** was obtained in 1.17 g (87% yield) with ideal 90% ee under standard conditions. In addition, we also tested the feasibility of this protocol in the late-stage

**Fig. 5** Synthetic applicability. **a** Gram-scale synthesis of **3a**. **b** The feasibility of complex molecule. **c** The chirality transfer processes

modification of complex molecules in Fig. 5b. To our delight, the estrone derivative **6** was well tolerated and gave the desired product **7** in 94% yield with excellent stereochemical integrity and high diastereoselectivity (>20:1 dr).

To demonstrate the synthetic utility of our N-C axially chiral amination products, based on our continuing interest in developing remote C-H functionalization, herein we designed a challenging remote chiral transformation in Fig. 5c. Allylic alkylation is an important component of chemistry and enantioselective studies, and particularly those of allylic alcohols, rely heavily on transition metals[57–59]. The *para*-position of the aniline C-H allylic alkylation was not reported. Herein, the N-C axially chiral amination product **3a** was used to react with (*E*)-1,3-diphenylprop-2-en-1-ol, which was catalyzed by DPP (diphenyl phosphate) at −30 °C in DCE for 24 h. The desired product was obtained in 72% yield. After the removal of Boc and the N-C axial chirality, a remote chiral 1- to 8-position transformation was achieved in two steps, in 54% yield with 51% ee. Further enantioselective catalysis studies of this allylic alkylation will be reported by us.

## Discussion

We have designed a π−π interaction and dual H-bond concerted control strategy and developed the CPAs catalyzed direct enantioselective nonbiaryl N-C axially chiral amination of 2-naphthylamine derivatives with azodicarboxylates as amino sources in good yields and high enantioselectivities. This type of N-C atropisomers is stabilized by intramolecular hydrogen bond and a broad range of substrates was proved to be well tolerated. Moreover, these axially chiral products could be converted into derivatives containing new chiral centers. Further mechanistic studies and the applications of this enantioselective axially chiral amination methodology in the synthesis of complex nitrogen-containing products are currently underway.

## Methods

**General procedure**. A mixture of 2-naphthylamine derivatives **1** (0.1 mmol, 1 equiv) and **CPA6** (0.01 mmol, 0.1 equiv), in DCM: Et₂O = 7:3 (1 mL) was stirred at −70 °C for 30 min in a 10 mL glass vial (purged sealed with PTFE cap). Then azodicarboxylate (0.2 mmol, 2 equiv) was added, and the reaction mixture was stirred at −70 °C for 48 h. The reaction mixture was direct purified by silica gel flash chromatography to give the amination product.

## Data availability

The data that support the findings of this study are available from the corresponding authors on request (DOI: 10.6084/m9.figshare.8131271). The X-ray crystallographic coordinates for structure reported in this article have been deposited at the Cambridge Crystallographic Data Centre (CCDC), under deposition numbers 1875011[https://www.ccdc.cam.ac.uk/structures/Search?Ccdcid=1875011&DatabaseToSearch=Published]. These data can be obtained free of charge from The Cambridge Crystallographic Data Centre via www.ccdc.cam.ac.uk/data_request/cif.

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

## Acknowledgements

This work was supported by the "Thousand Youth Talents Plan", NSFC (21672145, 51733007, 21702136), the Shuguang program (16SG10) from SEDF & SMEC, the STCSM (17JC1403700), and The Drug Innovation Major Project of the Ministry of Science and Technology of China (2018ZX09711001-005-002). We thank Prof. Yong-Qiang Tu and Prof. Wan-bin Zhang (Shanghai Jiao Tong University) for helpful suggestions and comments on this manuscript.

## Author contributions

H.-Y.B. performed the experiments and analyzed the data. F.-X.T. and T.-Q.L. helped with characterizing some new compounds. G.-D.Z, J.-M.T., T.-M.D. and Z.-M.C. provided useful and valuable advice. S.-Y.Z. conceived and directed the project and wrote the paper.

## Additional information

**Competing interests:** The authors declare no competing interests.

