## [Peer Review File · Nature Communications]

Reviewers' comments:

Reviewer #1 (Remarks to the Author):

In this communication, the group of Zhang describes a rare example of a successful chiral phosphoric acid activation of naphthylamine derivatives toward regioselective C-H amination by addition to azodicarboxylates for the efficient enantioselective synthesis of atropisomeric non-biaryl naphthalene-1,2-diamine compounds. This work is in continuation of the author's recent interest for the development of novel C-H amination methods but constitutes a highly important synthetic breakthrough in the still growing field of enantioselective organocatalysis for the control of non-central chirality that clearly deserves to be presented to the scientific community urgently.

This apparently simple approach constitutes the first direct catalyst-controlled enantioselective N-C axially chiral amination of 2-naphthylamines allowing a very original one-step synthesis of an important new class of axially chiral 1,2-diamines with good yields and excellent enantioselectivities in most cases. A crucial pi-pi interaction and dual H-bond concerted stereocontrol is clearly evidenced by selected experiments and strongly supports the mechanistic proposal proceeding through a central-to-axial conversion of chirality.

The transformation is scalable without significant loss of efficiency and both the robustness and versatility of this unique methodology are demonstrated by the good substrate scope and the wide functional variability tolerated including complex molecular scaffolds such as 6. The synthetic utility of the axially chiral 1,2-diamino naphthalene products has been illustrated with a diastereoselective allylation proceeding with a moderate but surprising 1,8-remote chirality transfer.

Overall, I have really appreciated this very nice piece of well-performed synthetic work offering a new practical one-pot access to enantiomerically enriched polyfunctionalized N-C axially chiral diamino naphthalene derivatives of a wide interest not only in synthetic organic chemistry.

Therefore, I give my strong and enthusiastic support for publication of this contribution in Nature Communications almost as it stands, after addressing the following very minor revisions:

-I suggest changing the word "asymmetric" by the more appropriated "enantioselective" throughout the manuscript including captions of schemes and tables.

-Lines 66, 90, 92 and 95: intermediate instead of intermedia

- Line 65: space after nucleophiles
- Lines 97 and 99: a verb is missing in this sentence, please rephrase
- Line 135: space after 91%
- Line 176: the word conjugate is not relevant
- Line 178: I suggest adding as a reference the following recent contribution from this referee's group on rearomatization with central-to-axial chirality conversion: J. Am. Chem. Soc. 2017, 139, 2140.
- Figure 4: the bond at the fused position of the imino-naphthyl intermediate II is missing
- For homogeneity reason, please change er into ee for compounds 3a and 8 and also in the conclusion.

Refereed by: Prof. Jean RODRIGUEZ from Aix-Marseille University (France)

Reviewer #2 (Remarks to the Author):

The preparation of optically pure atropisomeric molecules has received increasing interest because the structural motif has prevalently appeared in very useful catalysts or ligands for the asymmetric catalysis. In this manuscript, Zhang and coworkers report a very interesting chiral phosphoric acid-catalyzed asymmetric amination of N-aryl-2-naphthylamines with azodicarboxylates. The results are promising and the authors showcase the great potential for further applications. As a limited number of catalytic atroposelective construction of nonbiaryl atropisomers are available, this work represents a significant advance in this field and will receive a broad readership. I would like to recommend the publication of the manuscript in Nature Communications after addressing following comments, suggestions, and questions:

1. Table 1, entries 7-8, the authors claimed that the π - π interaction was the reason to improve the enantioselectivity. However, the more steric hindrance that the phenyl substituent provides might be the real reason for the enhancement of the stereoselectivity. Thus, the authors is suggested to investigate some more protecting groups of the nitrogen, such as tert-butyl, etc to get a deep insight into the substituent effect on the stereoselectivity.

2. Page 4-5, Line 88-90, The sentence “we believe the main reason is that our amino source azodicarboxylate is acyclic, which has only the dual H-bond to control the contortion of the 10-member ring transition state of intermediate B, which still represents a very challenging task” reads weird and I don't understand what the acyclic means and why represents a very challenging task.
3. Table 3, DEAD, DIAD and dibenzyl azodicarboxylate gave the desired product 3b-3d in low enantioselectivities (20-67% ee). Why? Is it possible that the optical active product undergoes racemization at the reaction temperature? The half-life of racemization of these compounds is suggested to be reported.
4. The NMR (5f, 5g) and HPLC (4b, 5d, 5h, 5i, 5l, 5m) spectra of the isolated material look dirty. The quality should be improved.
5. There are grammatical/typographical errors, such as, “Page 3, Line 65, nucleophilesderivative”, “Page 4, Line 72, 2 naphthylamine”. The linguistic improvement is welcome.

Reviewer #3 (Remarks to the Author):

In this manuscript, Zhang and co-workers disclosed the chiral phosphoric acid catalyzed asymmetric C-H amination of 2-naphthylamine with azodicarboxylate as the nitrogen source. The new type of N-aryl atropisomers were provided in high yields with excellent enantioselectivities for most cases. The successful application in gram-scale reaction and complex substrate further extended the utility of this protocol. Besides, the axial chirality was able to induce the generation of the central chirality far away in remote manner with acceptable stereocontrol. Overview, it is very similar to Jørgensen's work reported in 2006, in which 2-naphthol nucleophiles were used as the substrates. The major advantage for this work is the stability of the product with the absence of C-8 substitution on the naphthalene ring. The probable intramolecular H-bond between the hydrogen of the amino group on the aromatic ring and the oxygen of the carbonyl group on the amide is proposed by the author to stabilize the chiral N-C axis. Nonetheless, it is still hard to explain the instability of the products in Jørgensen's work with 2-naphthol structure where there is a possible intramolecular H-bond too. Meanwhile, the experimental results of 3b-3d with less bulky substitution illustrated that the H-bond effect is one factor for the stereocontrol. So in my view, the steric effect of both t-butyl group of DBAD and phenyl of the protecting group is the main factor for the stability of the chiral axis. Moreover, the π - π interaction mainly discussed by the author as the other main novelty for this work

is also arguable without more direct evidences. Definitely, I cannot give my support on the publication of this work in Nature Communications.

Several serious issues listed below needed to be addressed.

- 1) In table 1, 4% ee and 7% ee were detected for the reaction with 1b and 1c as the substrates. The author needs to confirm the low enantioselectivity was originated from the racemization during the reaction process or the hard stereocontrol. Hence, the stability examination of all the products is quite necessary for this work.
- 2) Only phenyl type protecting groups were tested here. The experimental results utilizing other type protecting groups, such as benzyl, allyl or sterically hindered t-butyl should be provided.
- 3) Bifunctional catalytic mode via dual H-bond between the catalyst and substrates is a feature of chiral phosphoric acid. It is better to remove the overemphasized statements for this point in the reaction design.
- 4) "Product 5e was not obtained due to the regioselectivity of the competing functionalization at the C5-position". Did the author isolate the byproduct with highly active diamine substrate and confirm its structure? Otherwise, it is inappropriate to give this statement in my opinion.
- 5) More mechanistic investigations should be performed if the authors want to introduce $\pi\pi$ interaction in their design. In addition, the steric effect of the phenyl protecting group should be taken into discussion too for the excellent enantiocontrol aside from H-bond.

Selected format errors, typos and other issues are listed as follows:

- 1) Figure 1b, "Jørgensen" should be corrected to "Jørgensen".
- 2) Figure 1c, there is no "R" in this figure, replaced "R" with "R" or "tBu". Similar issue appears in Figure 3. Table 3, why the authors use "R", "R3" and "R4" in this table instead of "R", "R1" and "R2"?
- 3) Line 65, "nucleophilesderivative" needs edit.
- 4) Line 69, "asymmetric catalysts in symmetric reactions" is hard to understand.
- 5) Line 71, "successively" or "successfully"?
- 6) Line 72, "2 naphthylamine" should be "2-naphthylamine".
- 7) Table 1, entry 1, there is a phenyl protecting group for the amine in substrate 1a, why there is no $\pi\pi$ interaction?
- 8) Table 1, entry 8, "yes" should be "Yes" accordingly. A space should be added between "47%" and "ee".
- 9) Lines 88-90, this sentence is difficult to read.

- 10) Line 91, "installing" should be changed to "install" and "group" should be "groups".
- 11) Line 102, "control strategies" should be "control strategy".
- 12) In the reaction optimization of Table 2, the temperature significantly affected the chemical yield. This critical information should be pointed out in the main text. What does the "(D)" or "(E)" mean in the solvent column in Table 2?
- 13) Line 132, "provided" should be corrected to "provide".
- 14) Line 135, "99%ee" should be "99% ee".
- 15) Line 139, "more sterically hindered substituted group showed notably decreased reactivity" is wrong statement. For the products 4a, 4d and 4e, R3 is methyl group. There is no "more sterically hindered substituted group" accordingly. The difference is the substituted position. Same statement was appeared in line 146.
- 16) Line 155, "electron-withdraw group" should be "electron-withdrawing group".
- 17) Line 160, delete "dramatically".
- 18) Figure 5b, delete the space between "94" and "%". Figure 5c, please add a space between the number and the unit symbol. For instance: "24" and "h".
- 19) Reverence 22, "full stop" should be added after anilides. Same error occurs in reverence 35.
- 20) Reverence 57, line 371, no capital for the first character of "Centers".

There are also many similar format errors in the Supporting Information. The author should try to polish the manuscript and the supplementary documents and remove all these issues before the publication.

1. Reviewer 1:

In this communication, the group of Zhang describes a rare example of a successful chiral phosphoric acid activation of naphthylamine derivatives toward regioselective C-H amination by addition to azodicarboxylates for the efficient enantioselective synthesis of atropisomeric non-biaryl naphthalene-1,2-diamine compounds. This work is in continuation of the author's recent interest for the development of novel C-H amination methods but constitutes a highly important synthetic breakthrough in the still growing field of enantioselective organocatalysis for the control of non-central chirality that clearly deserves to be presented to the scientific community urgently.

This apparently simple approach constitutes the first direct catalyst-controlled enantioselective N-C axially chiral amination of 2-naphthylamines allowing a very original one-step synthesis of an important new class of axially chiral 1,2-diamines with good yields and excellent enantioselectivities in most cases. A crucial pi-pi interaction and dual H-bond concerted stereocontrol is clearly evidenced by selected experiments and strongly supports the mechanistic proposal proceeding through a central-to-axial conversion of chirality.

The transformation is scalable without significant loss of efficiency and both the robustness and versatility of this unique methodology are demonstrated by the good substrate scope and the wide functional variability tolerated including complex molecular scaffolds such as **6**. The synthetic utility of the axially chiral 1,2-diamino naphthalene products has been illustrated with a diastereoselective allylation proceeding with a moderate but surprising 1,8-remote chirality transfer.

Overall, I have really appreciated this very nice piece of well-performed synthetic work offering a new practical one-pot access to enantiomerically enriched polyfunctionalized N-C axially chiral diamino naphthalene derivatives of a wide interest not only in synthetic organic chemistry.

Therefore, I give my strong and enthusiastic support for publication of this contribution in Nature Communications almost as it stands, after addressing the following very minor revisions:

Recommended modifications:

- 1) "I suggest changing the word "asymmetric" by the more appropriated "enantioselective" throughout the manuscript including captions of schemes and tables."

Response: We change "asymmetric" to "enantioselective" and highlighted in the revised manuscript.

- 2) "Lines 66, 90, 92 and 95: intermediate instead of intermedia."

Response: We change "intermedia" to "intermediate" and highlighted in the revised manuscript.

- 3) "Line 65: space after nucleophiles"

Response: This suggestion has been addressed in the revised manuscript.

- 4) "Lines 97 and 99: a verb is missing in this sentence, please rephrase."

Response: We changed this sentence, and the new sentence is "To evaluate the efficiency of our modified design, we used N-phenyl substituted 2-naphthylamine **1d** to react with DBAD in the presence of phenyl-substituted **CPA3** as a catalyst" and highlighted in the revised manuscript.

- 5) "Line 135: space after 91%."

Response: This suggestion has been addressed in the revised manuscript.

- 6) "Line 176: the word conjugate is not relevant."

Response: We change "conjugate" to "nucleophilic" and highlighted in the revised manuscript.

- 7) "Line 178: I suggest adding as a reference the following recent contribution from this referee's group on rearomatization with central-to-axial chirality conversion: J. Am. Chem. Soc. 2017, 139, 2140."

Response: Thank you for this suggestion. We added this reference as "reference 55" and highlighted in the revised manuscript.

- 8) "Figure 4: the bond at the fused position of the imino-naphthyl intermediate **II** is missing."

Response: We added the bond in the revised manuscript.

- 9) "For homogeneity reason, please change er into ee for compounds **3a** and **8** and also in the conclusion."

Response: We change "er" to "ee" in Fig. 5 (now Fig. 6) and conclusion, and highlighted in the revised

2. Reviewer 2:

The preparation of optically pure atropisomeric molecules has received increasing interest because the structural motif has prevalently appeared in very useful catalysts or ligands for the asymmetric catalysis. In this manuscript, Zhang and coworkers report a very interesting chiral phosphoric acid-catalyzed asymmetric amination of N-aryl-2-naphthylamines with azodicarboxylates. The results are promising and the authors showcase the great potential for further applications. As a limited number of catalytic atroposelective construction of nonbiaryl atropisomers are available, this work represents a significant advance in this field and will receive a broad readership. I would like to recommend the publication of the manuscript in Nature Communications after addressing following comments, suggestions, and questions:

Recommended modifications:

- 1) “Table 1, entries 7-8, the authors claimed that the π - π interaction was the reason to improve the enantioselectivity. However, the more steric hindrance that the phenyl substituent provides might be the real reason for the enhancement of the stereoselectivity. Thus, the authors is suggested to investigate some more protecting groups of the nitrogen, such as tert-butyl, etc to get a deep insight into the substituent effect on the stereoselectivity.”

Response: We thank the reviewer for these very good suggestions. In this revised manuscript, more N-substituted 2-naphthylamines bearing tert-butyl, benzyl, and allyl were examined and added in Table 1, entries 9-11. These substrates showed lower efficiency (19-37% yields) and low level of ee (0-9%). Notably, the more steric hindrance substrate N-*tert*-butyl-2-naphthylamines just gave 23% yield with 0 % ee. This result also supports that the effect of π - π interaction is the main reason for the enhancement of the enantioselectivity in this reaction.

- 2) “Page 4-5, Line 88-90, The sentence “we believe the main reason is that our amino source azodicarboxylate is acyclic, which has only the dual H-bond to control the contortion of the 10-member ring transition state of intermediate B, which still represents a very challenging task” reads weird and I don’t understand what the acyclic means and why represents a very challenging task.”

Response: Thank you for this comment. We rewrote this sentence and the new statement was “we believe the main reason is that the rotation is unhindered using only the dual H-bond to control the transition state of intermediate **B**.” highlighted in the revised manuscript.

- 3) “Table 3, DEAD, DIAD and dibenzyl azodicarboxylate gave the desired product 3b-3d in low enantioselectivities (20-67% ee). Why? Is it possible that the optical active product undergoes racemization at the reaction temperature? The half-life of racemization of these compounds is suggested to be reported.”

Response: We thank the reviewer for these very good questions. Based on these suggestions, we added a new part as “**Stereo-stability study**” in our revised manuscript. In this part, we studied the reaction process carefully and tested the half-lives of racemization of our N-C atroposelective products. These

results indicated that the N-C nonbiaryl atropisomers were stable at reaction conditions without racemization, and the low enantioselectivities mainly originated from the hard stereocontrol. More details please see our revised manuscript.

- 4) “The NMR (5f, 5g) and HPLC (4b, 5d, 5h, 5i, 5l, 5m) spectra of the isolated material look dirty. The quality should be improved.”

Response: Thank you for this comment. We redid those NMR spectra in 700 MHz and redid HPLC spectra using a new HPLC in the revised SI.

- 5) “There are grammatical/typographical errors, such as, “Page 3, Line 65, nucleophilesderivative”, “Page 4, Line 72, 2 naphthylamine”. The linguistic improvement is welcome.”

Response: We corrected the errors and made some linguistic improvement highlighted in the revised manuscript.

Reviewer 3:

In this manuscript, Zhang and co-workers disclosed the chiral phosphoric acid catalyzed asymmetric C-H amination of 2-naphthylamine with azodicarboxylate as the nitrogen source. The new type of N-aryl atropisomers were provided in high yields with excellent enantioselectivities for most cases. The successful application in gram-scale reaction and complex substrate further extended the utility of this protocol. Besides, the axial chirality was able to induce the generation of the central chirality far away in remote manner with acceptable stereocontrol. Overview, it is very similar to Jørgensen’s work reported in 2006, in which 2-naphthol nucleophiles were used as the substrates. The major advantage for this work is the stability of the product with the absence of C-8 substitution on the naphthalene ring. The probable intramolecular H-bond between the hydrogen of the amino group on the aromatic ring and the oxygen of the carbonyl group on the amide is proposed by the author to stabilize the chiral N-C axis.

Nonetheless, it is still hard to explain the instability of the products in Jørgensen’s work with 2-naphthol structure where there is a possible intramolecular H-bond too.

Response: We checked the X-ray file of Jørgensen’s report (ref.35). Notably, 4.399 Å between the OH and the oxygen of the carbonyl group could not form the H-bond.

Meanwhile, the experimental results of 3b-3d with less bulky substitution illustrated that the H-bond effect is one factor for the stereocontrol. So in my view, the steric effect of both t-butyl group of DBAD and phenyl of the protecting group is the main factor for the stability of the chiral axis. Moreover, the π - π interaction mainly discussed by the author as the other main novelty for this work is also arguable without more direct evidences. Definitely, I cannot give my support on the publication of this work in Nature Communications.

Response: In this revised manuscript, more N-substituted 2-naphthylamines bearing *tert*-butyl, benzyl, and allyl were examined and added in Table 1, entries 9-11. These result supports that the effect of π - π interaction is the main reason for the enhancement of the enantioselectivity in this reaction. In addition, we added a new part as “**Stereo-stability study**” in our revised manuscript. These results and discusses indicated that the intramolecular hydrogen bond (N-H \cdots O) notably improves the stereo-stability of these N-C axially chiral products. More details please see our revised manuscript.

Recommended modifications:

- 1) “In table 1, 4% ee and 7% ee were detected for the reaction with **1b** and **1c** as the substrates. The author needs to confirm the low enantioselectivity was originated from the racemization during the reaction process or the hard stereocontrol. Hence, the stability examination of all the products is quite necessary for this work.”

Response: We thank the reviewer for these very good questions. Based on these suggestions. We added a new part as “**Stereo-stability study**” in our revised manuscript. In this part, we studied the reaction process carefully and tested the half-life of racemization of our N-C atroposelective products. These results indicated that the N-C nonbiaryl atropisomers were stable at reaction conditions without racemization, the low enantioselectivities mainly originated from the hard stereocontrol. More details please see our revised manuscript. In addition, we also did three parallel experiments for **1b** as the substrate, and tested the ee values at 30 min (4.1 % ee), 90 min (3.7 % ee) and 5 h (4.6 % ee), showing that the N-C nonbiaryl atropisomers were stable at chiral reaction conditions.

- 2) “Only phenyl type protecting groups were tested here. The experimental results utilizing other type protecting groups, such as benzyl, allyl or sterically hindered t-butyl should be provided.”

Response: We thank the reviewer for these very good suggestions. In this revised manuscript, more N-substituted 2-naphthylamines bearing tert-butyl, benzyl, and allyl were examined and added in Table 1, entries 9-11. These substrates showed lower efficiency (19-37% yields) and low level of ee (0-9%). Notably, the more steric hindrance substrate N-*tert*-butyl-2-naphthylamines just gave 23% yield with 0 % ee. This result also supports that the effect of π - π interaction is the main reason for the enhancement of the enantioselectivity in this reaction.

- 3) “Bifunctional catalytic mode via dual H-bond between the catalyst and substrates is a feature of chiral phosphoric acid. It is better to remove the overemphasized statements for this point in the reaction design.”

Response: These suggestions have been addressed in figure 2 and our revised manuscript.

- 4) “Product **5e** was not obtained due to the regioselectivity of the competing functionalization at the C5-position”. Did the author isolate the byproduct with highly active diamine substrate and confirm its structure? Otherwise, it is inappropriate to give this statement in my opinion.”

Response: Thank you for this comment. In order to be more rigorous, we rewrote this sentence and the new statement was “the reaction was very complex and the desired C-1 amination product **5e** was not obtained” and highlighted in the revised manuscript.

- 5) “More mechanistic investigations should be performed if the authors want to introduce π - π interaction in their design. In addition, the steric effect of the phenyl protecting group should be taken into discussion too for the excellent enantiocontrol aside from H-bond.”

Response: We thank the reviewer for these good suggestions. In order to confirm the π - π interaction, in this revised manuscript, more N-substituted 2-naphthylamines bearing *tert*-butyl, benzyl, and allyl were examined and added in Table 1, entries 9-11. These substrates showed lower efficiency (19-37% yields) and low level of ee (0-9%). Notably, the more steric hindrance substrate N-*tert*-butyl-2-naphthylamines

just gave 23% yield with 0 % ee, this result also supports that the effect of π - π interaction is the main reason for the enhancement of the enantioselectivity in this reaction. Furthermore, we added a new part as “**Stereo-stability study**” in our revised manuscript. In this part, we studied the reaction process carefully and tested the half-life of racemization of our N-C atroposelective products. These results and discussions indicated that the intramolecular hydrogen bond (N-H \cdots O) notably improves the stereo-stability of these N-C axially chiral products. More details please see our revised manuscript.

Selected format errors, typos and other issues are listed as follows:

Comment #1: “Figure 1b, “JØrgensen” should be corrected to “Jørgensen.”

Response: We have changed “JØrgensen” to “Jørgensen” in Figure 1b in our revised manuscript.

Comment #2: “Figure 1c, there is no “R” in this figure, replaced “R” with “R” or “tBu”. Similar issue appears in Figure 3. Table 3, why the authors use “R”, “R3” and “R4” in this table instead of “R”, “R1” and “R2”?”

Response: These suggestions have been addressed in our revised manuscript. We changed “R” to “*t*-Bu” in Figure 1c. The Figure 2, Figure 3, Table 1, Table 2 and Table 3, we regarded them as an indivisible whole to demonstrate our experimental scheme. In Figure 2, there was “R”, and in Table 1, there were “R¹” and “R²”. In order to better describe the whole, so we numbered them as “R”, “R³” and “R⁴”.

Comment #3: “Line 65, “nucleophilesderivative” needs edit.”

Response: These suggestions have been addressed in our revised manuscript.

Comment #4: “Line 69, “asymmetric catalysts in symmetric reactions” is hard to understand.”

Response: We changed “asymmetric catalysts in symmetric reactions” to “enantioselective catalysts in various reactions” highlighted in the revised manuscript.

Comment #5: “Line 71, “successively” or “successfully?”

Response: We changed “successively” to “successfully” highlighted in the revised manuscript.

Comment #6: “Line 72, “2 naphthylamine” should be “2-naphthylamine.”

Response: We changed “2 naphthylamine” to “2-naphthylamine” highlighted in the revised manuscript.

Comment #7: “Table 1, entry 1, there is a phenyl protecting group for the amine in substrate 1a, why there is no π - π interaction?”

Response: In Table 1, entry 1, the substituent of CPA1 is H, and so there is no π - π interaction. There is π - π interaction between **1a** and CPA3 (Table 1, entry 3).

Comment #8: “Table 1, entry 8, “yes” should be “Yes” accordingly. A space should be added between “47%” and “ee”.”

Response: These suggestions have been addressed in our revised manuscript.

Comment #9: “Lines 88-90, this sentence is difficult to read.”

Response: We rewrote this sentence and the new statement was “we believe the main reason is that the

rotation is unhindered using only the dual H-bond to control the transition state of intermediate **B**.” highlighted in the revised manuscript.

Comment #10: “Line 91, “installing” should be changed to “install” and “group” should be “groups”

Response: We changed “installing” to “install” and changed “group” to “groups” highlighted in the revised manuscript.

Comment #11: “Line 102, “control strategies” should be “control strategy”.”

Response: We changed “control strategies” to “control strategy” highlighted in the revised manuscript.

Comment #12: “In the reaction optimization of Table 2, the temperature significantly affected the chemical yield. This critical information should be pointed out in the main text. What does the “(D)” or “(E)” mean in the solvent column in Table 2?”

Response: We added the temperature information “lowering the temperature would decrease the chemical yield (entry 13)” in the main text and highlighted in the revised manuscript. The “D” is short for DCM (Table 2, entry 1) and the “E” is short for Et₂O (Table 2, entry 8). We used the abbreviations for the mixed solvent.

Comment #13: “Line 132, “provided” should be corrected to “provide”.

Response: We changed “provided” to “provide” highlighted in the revised manuscript.

Comment #14: “Line 135, “99%ee” should be “99% ee”

Response: These suggestions have been addressed in our revised manuscript.

Comment #15: “Line 139, “more sterically hindered substituted group showed notably decreased reactivity” is wrong statement. For the products 4a, 4d and 4e, R₃ is methyl group. There is no “more sterically hindered substituted group” accordingly. The difference is the substituted position. Same statement was appeared in line 146.”

Response: We rewrote this sentence in line 139 and the new statement was “In addition, when the substituent was on meta-position or ortho-position, the reactivity was notably decreased.” in the revised manuscript.

We rewrote the sentence in line 146 and the new statement was “Using methyl as a substituent at C3 position, the reactivity was notably decreased but the stereoselectivity (5d) was slightly affected.” in the revised manuscript.

Comment #16: “Line 155, “electron-withdraw group” should be “electron-withdrawing group”

Response: We changed “electron-withdraw group” to “electron-withdrawing group” highlighted in the revised manuscript.

Comment #17: “Line 160, delete “dramatically”

Response: We deleted “dramatically” in the revised manuscript.

Comment #18: “Figure 5b, delete the space between “94” and “%”. Figure 5c, please add a space between the number and the unit symbol. For instance: “24” and “h”

Response: These suggestions have been addressed in our revised manuscript.

Comment #19: “Reverence 22, “full stop” should be added after anilides. Same error occurs in reverence 35.”

Response: We added “full stop” after anilides in reference 22 and added “full stop” after amination in reference 35 highlighted in the revised manuscript.

Comment #20: “Reverence 57, line 371, no capital for the first character of “Centers”.”

Response: We changed “Centers” to “centers” in reference 57 (now 58) highlighted in the revised manuscript.

REVIEWERS' COMMENTS:

Reviewer #2 (Remarks to the Author):

The revised manuscript has completely addressed my concerns and thus merits the publication. The chiral phosphoric acid-catalyzed asymmetric amination of N-aryl-2-naphthlamines with azodicarboxylates is really wonderful and will draw a great deal of attention in the future. I am very happy to recommend the publication of the manuscript in Nature Communications without reservation.

Reviewer #3 (Remarks to the Author):

In my point of view, the major weakness for this work is the lack of the sufficient novelty and creativity. First, the construction of N-C atropisomeric frameworks through an asymmetric Friedel-Crafts amination by mean of organocatalysis has been reported by Jørgensen for more than 10 years (*Angew. Chem. Int. Ed.* 2006, 45, 1147). The strategy here is almost the same as Jørgensen's report. The unique bifunctional catalytic mode and π - π interaction are well-known features of CPA, it is not suitable for the author to claim that these features represent the significant creativity of this work. Next, intramolecular H-bond assisted enhancement of stability of N-C axis has also been investigated (*J. Am. Chem. Soc.* 2009, 131, 54; *J. Org. Chem.* 2010, 75, 5031). Furthermore, the applicability of the obtained structures has never been verified.

Leave aside the insights of the mechanism; some other key points are still ambiguous even in the revised manuscript. For instance, the stability of products 3b-3d has been raised by two reviewers, however, related stability examination for these compounds remain missing. Obviously, the lower enantioselectivities were mainly attributed to the less steric substitutions for these cases. It is also inappropriate for the statement that "our N-C axially chiral products have suitable stability at room temperature" when most of the products gave about only 20-40 hours racemization half-lives. The remarkable difference between 5g (221.9 days) and other products clearly illustrated that the steric effect, particularly in C-8 position of naphthalene ring is another essential factor for the stability of the chiral axis.

I have no doubt the effect of π - π interaction in the stereocontrol and H-bond in the stability of the products. However, these features are still insufficient to support this work for publication on Nature Communications. Therefore, I prefer to adhere to my previous opinion.

Reviewer #2 (Remarks to the Author):

The revised manuscript has completely addressed my concerns and thus merits the publication. The chiral phosphoric acid-catalyzed asymmetric amination of N-aryl-2-naphthlamines with azodicarboxylates is really wonderful and will draw a great deal of attention in the future. I am very happy to recommend the publication of the manuscript in Nature Communications without reservation.

Reviewer #3 (Remarks to the Author):

In my point of view, the major weakness for this work is the lack of the sufficient novelty and creativity. First, the construction of N-C atropisomeric frameworks through an asymmetric Friedel–Crafts amination by mean of organocatalysis has been reported by Jørgensen for more than 10 years (Angew. Chem. Int. Ed. 2006, 45, 1147). The strategy here is almost the same as Jørgensen’s report. The unique bifunctional catalytic mode and π - π interaction are well-known features of CPA, it is not suitable for the author to claim that these features represent the significant creativity of this work. Next, intramolecular H-bond assisted enhancement of stability of N-C axis has also been investigated (J. Am. Chem. Soc. 2009, 131, 54; J. Org. Chem. 2010, 75, 5031). Furthermore, the applicability of the obtained structures has never been verified. Leave aside the insights of the mechanism; some other key points are still ambiguous even in the revised manuscript. For instance, the stability of products 3b-3d has been raised by two reviewers, however, related stability examination for these compounds remain missing. Obviously, the lower enantioselectivities were mainly attributed to the less steric substitutions for these cases. It is also inappropriate for the statement that “our N-C axially chiral products have suitable stability at room temperature” when most of the products gave about only 20-40 hours racemization half-lives. The remarkable difference between 5g (221.9 days) and other products clearly illustrated that the steric effect, particularly in C-8 position of naphthalene ring is another essential factor for the stability of the chiral axis. I have no doubt the effect of π - π interaction in the stereocontrol and H-bond in the stability of the products. However, these features are still insufficient to support this work for publication on Nature Communications. Therefore, I prefer to adhere to my previous opinion.

Responses:

In my point of view, the major weakness for this work is the lack of the sufficient novelty and creativity. First, the construction of N-C atropisomeric frameworks through an asymmetric Friedel–Crafts amination by mean of organocatalysis has been reported by Jørgensen for more than 10 years (Angew. Chem. Int. Ed. 2006, 45, 1147). The strategy here is almost the same as Jørgensen’s report. The unique bifunctional catalytic mode and π - π interaction are well-known features of CPA, it is not suitable for the author to claim that these features represent the significant creativity of this work.

Response: Although Jørgensen has reported his asymmetric Friedel–Crafts amination, our work is very different from that in substrates, catalysts and strategy.

1. The 2-naphthol derivatives used by Jørgensen have been widely used in asymmetric reactions for decades, while the enantioselective construction of 2-naphthylamine derivatives is rare and the more

challenging chiral nonbiaryl N-C atropisomers of 2-naphthylamine derivatives have not been discovered.

2. We use chiral phosphoric acid as catalyst and the acidic catalytic mode is completely different from the alkaline catalytic system of cinchona alkaloid.
3. What's important is the π - π interaction and dual H-bond concerted control strategy, which is of significant novelty and creativity. The bifunctional catalytic mode is well-known feature of CPA, but it could not control the enantioselectivity of our reaction. In our manuscript, the concerted control plays a decisive role in stereoselectivity, and it is novel.

Furthermore, the applicability of the obtained structures has never been verified.

Response: To enhance the applicability, the gram-scale reaction, the feasibility in the late-stage modification of complex molecules and the challenging remote chiral transformation of the *para*-position of the aniline C-H allylic alkylation were carried out. More applications of the obtained structures, such as chiral ligands, are in progress and will be reported in the following papers.

Leave aside the insights of the mechanism; some other key points are still ambiguous even in the revised manuscript. For instance, the stability of products **3b-3d** has been raised by two reviewers, however, related stability examination for these compounds remain missing. Obviously, the lower enantioselectivities were mainly attributed to the less steric substitutions for these cases.

Response: Actually, we answered this question to the reviewer in the last manuscript (NCOMMS-19-05357A) and last point-by-point response, and the reviewer #2 also said "revised manuscript has completely addressed my concerns".

In the last point-by-point response to the question 3 of reviewer #2, we answered that "We thank the reviewer for these very good questions. Based on these suggestions, we added a new part as "Stereo-stability study" in our revised manuscript. In this part, we studied the reaction process carefully and tested the half-lives of racemization of our N-C atroposelective products. These results indicated that the N-C nonbiaryl atropisomers were stable at reaction conditions without racemization, and the low enantioselectivities mainly originated from the hard stereocontrol. More details please see our revised manuscript."

In the last revised manuscript (NCOMMS-19-05357A), we added the contents "Based our above experiment results (**3a-3d**), we knew that the more steric hindrance of azodicarboxylates gave the better enantioselectivity. Furthermore, in order to confirm that the low enantioselectivity was originated from the racemization during the reaction process or the hard stereo-control, we did three parallel experiments to obtain the **3d** at 12 h (64 % ee), 24 h (66 % ee) and 48 h (67 % ee) under standard conditions, showing that the N-C nonbiaryl atropisomers were stable at reaction conditions without racemization. Compared **3a** (92% ee, 70.9 h) and **3d** (67% ee, 25.0 h), this evidence suggested that the large steric hindrance of azodicarboxylates (Fig.5, part I) not only improved the enantioselectivity of reaction, but also stabilized the N-C atropisomers."

The ee values of **3b** (23%) and **3c** (20%) were low and might have big errors to examine the half-lives of racemization, so we used **3a** and **3d** as examples and detailedly tested the relationship between the stability and the steric hindrance. The conclusion is obvious and accurate.

Next, intramolecular H-bond assisted enhancement of stability of N-C axis has also been investigated (J. Am. Chem. Soc. 2009, 131, 54; J. Org. Chem. 2010, 75, 5031). It is also inappropriate for the statement that "our N-C axially chiral products have suitable stability at room temperature" when most of the

products gave about only 20-40 hours racemization half-lives. The remarkable difference between **5g** (221.9 days) and other products clearly illustrated that the steric effect, particularly in C-8 position of naphthalene ring is another essential factor for the stability of the chiral axis.

Response: The N-C atropisomers stabilized by intramolecular H-bond are very rare and challenging structures, and worth further exploration. As the reviewer 3 mentions, the few reports (*J. Am. Chem. Soc.* **2009**, 131, 54; *J. Org. Chem.* **2010**, 75, 5031) introduce that the half-life of racemization is just 10 s without the help of sterically substituents.

Our structures are stabilized by intramolecular H-bonds, and it's enhanced greatly with 20-100 hours racemization half-lives compared to previous reports. Furthermore, using **3a** (90.10% ee) as an example, the half-life of racemization is 1048 hours (43.7 days) at 5 °C in *n*-hexane, and 5807 hours (242 days) at -18 °C in *n*-hexane. These results indicate that our N-C axially chiral products can be used in various asymmetric reactions. At solid state, it is 90.04% ee after 90 h at -18 °C and so the products can be easily stored. The new experimental results and references (refs 56 and 57) were added in the revised manuscript.